# The Health Economic Evaluation of Bariatric Surgery Versus a Community Weight Management Intervention Analysis from the Idiopathic Intracranial Hypertension Weight Trial (IIH:WT)

**DOI:** 10.3390/life11050409

**Published:** 2021-04-30

**Authors:** Magda Aguiar, Emma Frew, Susan P. Mollan, James L. Mitchell, Ryan S. Ottridge, Zerin Alimajstorovic, Andreas Yiangou, Rishi Singhal, Abd A. Tahrani, Alex J. Sinclair

**Affiliations:** 1Health Economics Unit, Institute of Applied Health Research, University of Birmingham, Birmingham B12 2TT, UK; magdaf.aguiar@gmail.com (M.A.); e.frew@bham.ac.uk (E.F.); 2Birmingham Neuro-Ophthalmology, University Hospitals Birmingham NHS Foundation Trust, Queen Elizabeth Hospital, Birmingham B15 2WB, UK; 3Institute of Metabolism and Systems Research, College of Medical and Dental Sciences, University of Birmingham, Birmingham B15 2TT, UK; j.mitchell.1@bham.ac.uk (J.L.M.); Z.Alimajstorovic@bham.ac.uk (Z.A.); A.Yiangou@bham.ac.uk (A.Y.); A.A.Tahrani@bham.ac.uk (A.A.T.); a.b.sinclair@bham.ac.uk (A.J.S.); 4Department of Neurology, University Hospitals Birmingham NHS Foundation Trust, Queen Elizabeth Hospital, Birmingham B15 2WB, UK; 5Birmingham Clinical Trials Unit, College of Medical and Dental Sciences, University of Birmingham, Birmingham B15 2TT, UK; r.ottridge@bham.ac.uk; 6Upper GI Unit and Minimally Invasive Unit, Birmingham Heartlands Hospital, University Hospitals Birmingham NHS Foundation Trust, Birmingham B9 5SS, UK; rishi.singhal@heartofengland.nhs.uk; 7Institute of Cancer and Genomic Sciences, College of Medical and Dental Sciences, University of Birmingham, Birmingham B15 2TT, UK; 8Centre for Endocrinology, Diabetes and Metabolism, Birmingham Health Partners, Birmingham B15 2GW, UK; 9Department of Endocrinology, University Hospitals Birmingham NHS Foundation Trust, Queen Elizabeth Hospital, Birmingham B15 2WB, UK

**Keywords:** bariatric surgery, cost-effectiveness, headache, health care system, intracranial pressure, national health service, obesity, papilloedema, pseudotumor cerebri, randomised control trial

## Abstract

Background: The Idiopathic Intracranial Hypertension Weight Trial (IIH:WT) established the efficacy of bariatric surgery as compared to a community weight management intervention in reducing intracranial pressure in active IIH. The aim of this cost-effectiveness analysis was to evaluate the economic impact of these weight loss treatments for IIH. Methods: IIH:WT was a five-year randomised, controlled, parallel group, multicentre trial in the United Kingdom, where participants with active IIH and a body mass index ≥35 kg/m^2^ were randomly assigned (1:1) to receive access to bariatric surgery or a community weight management intervention. All clinical and quality of life data was recorded at baseline, 12 and 24 months. Economic evaluation was performed to assess health-care costs and cost-effectiveness. Evaluations were established on an intention to treat principle, followed by a sensitivity analysis using a per protocol analysis. Results: The mean total health care costs were GBP 1353 for the community weight management arm and GBP 5400 for the bariatric surgery arm over 24 months. The majority of costs for the bariatric surgery arm relate to the surgical procedure itself. The 85% who underwent bariatric surgery achieved a 12.5% reduction in intracranial pressure at 24 months as compared to 39% in the community weight management arm; a mean difference of 45% in favour of bariatric surgery. The cost effectiveness of bariatric surgery improved over time. Conclusions: The IIH:WT was the first to compare the efficacy and cost-effectiveness of bariatric surgery with community weight management interventions in the setting of a randomised control trial. The cost-effectiveness of bariatric surgery improved over time and therefore the incremental cost of surgery when offset against the incremental reduction of intracranial pressure improved after 24 months, as compared with 12 months follow up.

## 1. Introduction

Idiopathic intracranial hypertension (IIH) causes challenging headaches, visual loss, and reduced quality of life [1]. Hospital costs for IIH admissions have been found to be four times greater than for a general population-based per person admission [2]. Both in the United Kingdom and the USA patients access care through the emergency room, and have multiple attendances [3,4]. The inappropriate use of emergency care and repeated disease recurrences [5,6] may reflect the lack of accessible disease modifying treatments [7]. These factors coupled with the increased incidence and prevalence of IIH [3,8,9] are driving rising health care costs [2,3].

Body weight is the main modifiable risk factor in IIH [10,11], specifically central adiposity [12]. Understanding this role of weight management in the disease was set as a high research priority by physicians, carers, and patients [13]. Guidelines for treatment of IIH have placed weight management central to modify the disease [14,15,16]. However, prior to the IIH weight trial (IIH:WT) there was little evidence for the best method of weight loss to achieve sustained IIH remission [17]. IIH:WT showed that bariatric surgery was superior to community weight management intervention in women with active IIH and a body mass index (BMI) ≥ 35 kg/m^2^ in lowering intracranial pressure, improved headache outcomes, and providing improvements in quality of life [18]. Sustained weight loss should prevent multiple relapses of the disease [5,6], which will translate into a substantial reduction in healthcare costs.

Herein we estimate the cost-effectiveness of bariatric surgery, as compared to a community weight management intervention, in the first 24 months following the intervention in IIH. The aim was to establish the economic value of weight management type within the IIH:WT.

## 2. Materials and Methods

Sixty six women were recruited to the multicentre randomised controlled trial IIH:WT, comparing the efficacy of a bariatric surgery pathway versus a community weight management intervention, Weight Watchers^TM^. Participants were identified from neurology and ophthalmology clinics at seven UK National Health Service (NHS) hospitals and recruited formally from five NHS centres. All participants had active IIH with papilloedema, in accordance with agreed criteria for diagnosis of IIH. The trial was registered, clinicaltrials.gov identifier: NCT02124486. The protocol and eligibility criteria [17] and the main study findings have previously been published [18].

This economic evaluation was undertaken from a UK National Health Service (NHS) and Personal Social Service perspective and measured costs and outcomes after 12 and 24 months follow up. The cost-effectiveness analysis compared the incremental costs and incremental reduction in intracranial pressure (ICP) for the bariatric surgery versus the community weight management pathway, as a treatment for IIH. A 12.5% reduction in ICP was chosen as the outcome for the cost-effectiveness analysis as this equated to the difference predicted between the trial arms. The incremental cost-effectiveness ratio was the difference in costs offset against the difference in ICP expressed as cost per reduction in ICP of 12.5%. A secondary per-protocol cost-effectiveness analysis was conducted.

### 2.1. Resource Use and Costs

To estimate costs, resource use data were collected and combined with unit cost information, including all primary care visits (General Practitioner and Nurse); hospital inpatient stays; hospital outpatient visits; Accident and Emergency visits; and all prescription medication. Primary care resource use was obtained from study-specific questionnaires and completed by trial participants at the trial data collection points. Secondary resource use and prescription data were obtained from the trial case report forms.

Unit costs for all primary and secondary resource use were obtained from the NHS Reference costs [19] and Unit Cost of Health and Social Care [20]; and for prescription costs, obtained from the British National Formulary [21]. For prescription costs, all drugs used to manage IIH related symptoms were included, and costs estimated by combining the unit price by the daily dosage and number of days of treatment. For the intervention costs, the cost of the weight management programme was obtained from the trial data and for surgery, the NHS reference cost was applied. All inpatient stays up to eleven bed-days were priced at the appropriate flat rate, stays beyond that incurred a day rate for inpatient care. All costs were in UK pounds sterling using the 2017–2018 price level. Appendix A reports all unit costs applied.

### 2.2. Analyses

The economic evaluation measured costs and effects at the end of 24 months, so compared the difference in costs and the difference in outcomes for weight management versus surgery, for treatment of IIH. The primary analysis adopted an ‘intention to treat’ principle. Secondary analysis also measured cost-effectiveness consistent with a ‘per protocol’ principle as per the clinical effectiveness evaluation. All statistical tests set the significance levels at 2.5% with 95% confidence intervals calculated.

The data were examined for missingness and missing data were imputed separately for GP, nurse, inpatient, outpatient, and all medication costs. Multiple imputations were performed using a chained model with 60 iterations to account for the high proportion of missing data. Multiple imputation replaced each missing value with a set of m plausible values to generate 60 replacement values (m = 60) for each of the missing cells in these data sets, using multiple linear regression models [22]. The imputed variables were merged at 0, 12, and 24 months.

All costs and outcomes were discounted to present values at an annual rate of 3.5%, in accordance with the National Institute for Health and Care Excellence [23].

### 2.3. Sensitivity Analyses

To estimate the uncertainty around the resulting incremental cost-effectiveness ratio (ICER), a probabilistic sensitivity analysis was conducted. This was done using a non-parametric bootstrap analysis, with 5000 replicates, which empirically constructs the distribution of cost-effectiveness ratios by producing 5000 paired cost/ICP estimates. We considered using only complete cases (no missing data imputation) for a further sensitivity analysis, but this was disregarded due to the high level of missing data and the strong likelihood of bias.

All analyses were conducted using Stata version 15 (StataCorp LLC, College Station, TX, USA).

## 3. Results

Sixty-six women, with a mean age of 32.0 years (standard deviation (SD) ± 7.8, range 20–53 years), were included in the IIH:WT. The mean body mass index was 43.9 ± 7.0 kg/m^2^ ranging from 35.3 to 63.3 kg/m^2^. The predominant surgery was Roux-en-Y gastric bypass in 12 participants (44%), followed by gastric banding in 10 participants (37%) and laparoscopic sleeve gastrectomy in five participants (19%).The mean number of Weight Watchers^TM^ face-to-face sessions attended was 14.3 (SD 10.6), with 58% attending at least one session. The adjusted difference in intracranial pressure at 12 months between the two trial arms was −6.0 cmCSF (adjusted mean difference (95% CI): −6.0 (−9.5, −2.4); *p* = 0.001). The effect on intracranial pressure increased between 12 and 24 months with a mean difference between the two arms of −8.2 cmCSF (adjusted mean difference (95% CI): −8.2 (−12.2, −4.2); *p* < 0.001).

### 3.1. Resource Use and Costs

There were no statistically significant differences between the two study arms in terms of primary or secondary care resource use, for the complete cases (Table 1).

The degree of missing resource use data was balanced across the two study arms (Appendix A). After imputation, the total mean cost, including all primary, secondary, and medical-use costs, were GBP 1353 for the weight watchers arm and GBP 5400 for the bariatric surgery arm. The mean between-group cost difference was GBP 4107 (95% CI 3334–4880). (Table 2)

The majority of costs for the bariatric surgery arm relate to the surgical procedure itself. Interestingly the primary, secondary, and medication costs (not including intervention costs) were consistently higher in the community weight management arm, and this difference was greater at 24 months compared to 12 months.

### 3.2. Intracranial Pressure Outcome

The proportion of patients who achieved a 12.5% reduction in ICP at 24 months were 85% with bariatric surgery and 39% with Weight Watchers. This represents a mean difference of 45% in favour of bariatric surgery (95% CI: 24% to 66%) (Table 3).

### 3.3. Incremental Analysis

When the costs and ICP differences were combined, they show that at 24 months, bariatric surgery cost GBP 8807 to achieve a 12.5% reduction in ICP, when compared to a community weight management intervention programme (Table 4).

The cost-effectiveness plane presents 5000 jointly bootstrapped cost-ICP pairs distributed across four quadrants. Most of the pairs lie in the north-east quadrant indicating bariatric surgery to be more expensive and to lead to more gains in ICP reduction, relative to weight management. (Figure 1).

### 3.4. Sensitivity Analysis

A sensitivity analysis performed a per-protocol analysis as two patients who were assigned to the community weight intervention had bariatric surgery between 12 and 24 months. Additionally, six participants who were assigned to the bariatric surgery arm did not have surgery and were therefore considered in the weight management arm. This per-protocol analysis showed that the ICERs were stable to these alterations, as detailed in Table 5.

## 4. Discussion

The IIH:WT established that bariatric surgery was superior to a community weight management intervention in reduction of intracranial pressure, headache outcomes, and delivery of better quality of life in those with active IIH and a body mass index ≥35 kg/m^2^ [18]. This cost-effectiveness analysis reports on the economic value of bariatric surgery and details how the initial high cost of bariatric surgery is offset by these superior gains in intracranial pressure when compared to a weight management programme over 24 months. The incremental cost-effectiveness ratio at 24 months is less than at 12 months indicating that bariatric surgery appears to be more cost-effective with time in active IIH.

Bariatric surgery results in sustainable long term weight loss [24], and in the IIH:WT reduced the intracranial pressure to below that of the diagnosis level, i.e., induced remission [18]. The durability of bariatric surgery will therefore likely reduce the frequent relapses reported in IIH [5,6], which contribute to significant costly frequent hospital admissions over time [2,3,4].

Bariatric surgery is known to deliver wide-reaching health benefits, as compared with conservative medical methods for weight loss [24]. For example, Roux-en-Y gastric bypass surgery is associated with a reduced risk of cardiovascular disease when compared with routine care [25]. These cardiovascular improvements may be of additional benefit for those with IIH, as IIH is known to be associated with a twofold increased risk of cardiovascular outcomes [8]. This means that the cost-effectiveness results reported here represent conservative estimates as these wider cost-savings as a result of having surgery have not been included.

Roux-en-Y gastric bypass surgery has been evaluated as the most clinically effective weight loss intervention compared with other bariatric procedures and weight management programmes [26]. This cost-effective analysis did not differentiate between the three different types of bariatric surgery offered in IIH:WT. The sample size within each surgical sub-type group was small; therefore, the differential costs and effects were not included. There is evidence that laparoscopic gastric band is associated with lower procedure costs, but has a much higher rate of revisional surgery, as well as a smaller and less well-maintained effect on body weight than Roux-en-Y gastric bypass and laparoscopic sleeve gastrectomy [26]. However, the range and distribution of bariatric surgeries performed in the IIH:WT trial broadly reflect current practice in the UK health care system, and therefore the results are applicable to assessing bariatric surgery versus community weight management for treatment of IIH. Globally, more research is needed to fully evaluate the differential cost-effectiveness between bariatric surgery types.

This analysis was conducted from a health service perspective meaning that any out-of-pocket payments or indirect costs of IIH were not included. If these indirect costs such as days off work or time to travel to appointments to manage on-going IIH symptoms were included, then this would likely make bariatric surgery even more cost-saving. For example, in the USA total economic costs were estimated at exceeding USD 444 million in 2007 [2], as compared in the UK to GBP 9.2 million in 2002 rising to GBP 49.9 million in 2014 [12], when limited to health care costs alone. Likewise, the UK has a national health care system, and likely does not reflect other types of health care systems, in terms of relative costs for procedures and hospital attendances which are costed differently across private and insurance health care systems.

## 5. Conclusions

Bariatric surgery has been shown to be cost-effective [26], however access to bariatric surgery within the UK health care system, the National Health Service, remains limited with less than 0.002% of the potentially eligible adults having the surgery annually [27]. The data in this study and the original randomised control trial [18] suggest that improved access to bariatric surgery for women with active IIH with a body mass index ≥35 kg/m^2^ is likely to be cost effective, with improved savings over the longer term. Lifestyle weight management interventions currently remains the first line of treatment in patients with obesity and IIH, and if such treatment fail to achieve IIH remission, then referral to a bariatric surgery pathway should be offered.

## Figures and Tables

**Figure 1 life-11-00409-f001:**
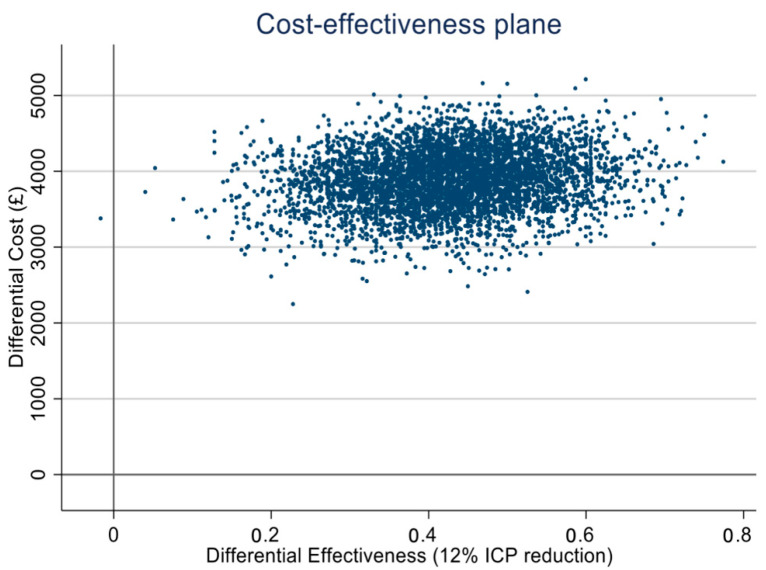
Cost-effectiveness plane 24 months (CEA).

**Table 1 life-11-00409-t001:** Resource use at 12 month and 24 months follow-up (complete case analysis). Note that where number of days is in hospital is noted the result is a portion of a day. Importantly with some forms of bariatric surgery, this is done as a day case and not an overnight/24 h stay in hospital. Likewise, not every person was seen in the outpatient department following recruitment to the trial, hence why this is not an integer.

***Resource Use at 12 Months***						
**Resource Use Item**	**Community Weight Management Intervention**		**Bariatric Surgery Pathway**		**Bootstrap Mean Difference**	**95% Confidence Interval (CI)**
	**Mean**	**SD**	**Mean**	**SD**		
Number of days in hospital	0.03 (n = 24)	0.17	0.12 (n = 29)	0.7	0.087	−0.15 to 0.33
Number of outpatient visits	0.21 (n = 24)	0.55	0.55	1.56	0.38	−0.24 to 1.01
Number of GP visits	4.6 (n = 27)	6.24	3.03 (n = 29)	4.62	−156	−4.57 to 1.4
Number of nurse contacts	0.46 (n = 24)	1.1	0.48 (n = 27)	1.34	−0.97	−6.85 to 8.81
Prescription medication	0.73 (n = 24)	1	0.15 (n = 29)	0.44	−0.58	−0.95 to −0.21
***Resource Use at 24 months***						
**Resource Use Item**	**Community Weight Management Intervention**		**Bariatric Surgery Pathway**		**Bootstrap Mean Difference**	**95% CI**
	**Mean**	**SD**	**Mean**	**SD**		
Number of days in hospital	0.67 (n = 22)	3.65	0.12 (n = 23)	0.55	−0.6	−1.97 to 0.78
Number of outpatient visits	0.42 (n = 22)	0.79	0.21 (n = 23)	0.89	−0.22	−0.62 to 0.17
Number of GP visits	6.0 (n = 22)	8.44	2.26 (n = 23)	5.5	−4.13	−0.815 to −0.11
Number of nurse contacts	0.53 (n = 19)	1.17	0.43 (n = 23)	1.31	−0.13	−9.17 to 8.91
Prescription medication	0.52 (n = 22)	0.67	0.09 (n = 24)	0.38	−0.42	−0.69 to −0.16

CI = confidence interval. GP = general practitioner. SD = standard deviation.

**Table 2 life-11-00409-t002:** Mean and incremental cost over 12 and 24 months (imputed).

Cost Item	Community Weight Management Intervention (*n* = 33)		Bariatric Surgery Pathway (*n* = 33)		Bootstrap Mean Difference	95% CI
	Mean (GBP)	SD	Mean (GBP)	SD		
**12 months**						
Secondary care costs	125.33	557.78	184.54	520	44.06	−243.56 to 331.7
Primary care costs	188.15	232.67	120.96	167.54	−67.83	−164.0 to 29.34
Prescription medications costs	116.53	278.5	33.42	117.66	−83.11	−185.67 to 19.46
Total cost	430.01	739.87	338.93	605.70	−29.05	−309.26 to 251.13
Total cost (including intervention costs)	624.01	739.88	5028.71	1675.31	4440.05	3837.52 to 5042.58
**24 months**						
Secondary care costs	402.26	1063.0	300.20	605.69	−145.94	−624.78 to 332.90
Primary care costs	593.44	693.01	326.30	373.30	−271.37	−538.85 to −3.89
Prescription medications costs	164.01	365.57	51.15	180.28	−112.86	−251.98 to 26.26
Total cost	1159.72	1435.61	677.65	878.66	−394.75	−922.24 to 132.77
Total cost (including intervention costs)	1353.72	1435.61	5400.61	1841.45	4107.19	3334.33 to 4880.05

CI = confidence interval. SD = standard deviation.

**Table 3 life-11-00409-t003:** Mean ICP and proportion of patients who achieved a 12% reduction of ICP at 12 and 24 months.

Cost item	Community Weight Management Intervention		Bariatric Surgery Pathway		Bootstrap Mean Difference	95% CI
	Mean	SD	Mean	SD		
**12 months**						
Intracranial pressure (cmCSF)	32.29	5.71	25.91	8.39	−6.48	−9.73 to −3.24
12% reduction in intracranial pressure	0.42	0.50	0.82	0.39	0.39	0.18 to 0.6
**24 months**						
Intracranial pressure (cmCSF)	30.89	5.17	23.65	7.4	−7.33	−10.2 to −4.42
12% reduction in intracranial pressure	0.39	0.5	0.85	0.36	0.45	0.24 to 0.66

CI = confidence interval. SD = standard deviation.

**Table 4 life-11-00409-t004:** Cost-effectiveness analysis (CEA).

	Incremental Costs (GBP)	Incremental Outcomes	Incremental Cost-Effectiveness Ratio (ICER)
**12 months**			
CEA	4440	0.39	11,181
mean (95% CI)	(3838 to 5043)	(0.18 to 0.6)	
**24 months** (costs and outcomes discounted at 3.5%)			
CEA	3963	0.45	8807
mean (95% CI)	(3229 to 4698)	(0.24 to 0.66)	

CEA = cost effectiveness analysis. CI = confidence interval. ICER= incremental cost-effectiveness ratio.

**Table 5 life-11-00409-t005:** Per protocol analysis.

	Incremental Costs	Incremental Outcomes	Incremental Cost-Effectiveness Ratio (ICER)
**12 months**			
CEA	4533	0.4	11,205
mean (95% CI)	(3877 to 5188)	(0.19 to 0.62)	
**24 months** (costs and outcomes discounted at 3.5%)			
CEA	4116	0.48	8570
mean (95% CI)	(3337 to 4895)	(0.29 to 0.68)	

CEA = cost effectiveness analysis. CI = confidence interval. ICER= incremental cost-effectiveness ratio.

## Data Availability

Reasonable requests will provide data beginning 12 months and ending 3 years after publication of this article to researchers whose proposed use of the data is approved by the original study investigators. Proposals should be made to the corresponding author and requesters will need to sign a data access agreement.

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
