# Peer review of "The Health Economic Evaluation of Bariatric Surgery Versus a Community Weight Management Intervention Analysis from the Idiopathic Intracranial Hypertension Weight Trial (IIH:WT)"

_life, 2021, doi:10.3390/life11050409_

Round 1
Reviewer 1 Report
The authors present the health economic analysis based on their two previously published studies comparing bariatric surgery with community weight loss in patients with idiopathic intracranial hypertension. This is a health priority for the management of IIH patients and their doctors. In certain health systems (such as the NHS) being able to provide economic data to back up a change in medical management protocols is vital. While this data may be of less direct applicability within a user-pays system like America or private health in Australia, it is still of interest.
Author Response
We would like to thank the reviewer for their expertise. We have made a note in the discussion regarding applicability of the research for different healthcare systems.
Reviewer 2 Report
this paper is interesting and it addresses a relevant problem and unmet needs within the treatment of IIH.
I could not find the paper corresponding to reference 18. I was interested on understanding how many centres were included in the trial.
the sample is rather small and there are less than 30 participants in each arm. however no mention is made regarding on how this might require some caution in the interpretation of the results. are all trials in this area this small? having missing data in such a small sample also raises questions on how results might apply to other settings. the reader is told about missing data and how this problem has been addressed but nothing is said about the percentage of missing data in specific variables or its trend over time.
Author Response
We would like to thank the reviewer for their considered opinion and helpful additions to our manuscript.
We have annotated the reference 18. It is published on 26th April at 11.00 eastern time.
We have added into the methods the number of NHS hospitals where the patients were recruited from.
This is a rare disease and the numbers were based on powering the primary outcome of intracranial pressure to detect a difference of more than 5 cm CSF, in fact we detected a difference of 6cm CSF at 12 months, achieving the primary outcome. So statistically the numbers are robust.
Supplemental table 2 gives the break down of the missing data at both 12 and 24 months. When performing an economic evaluation missing data at any timepoint needs to be handled, as it was done within this evaluation.